# Synthesis and Antimicrobial Activity of the Pathogenic *E. coli* Strains of *p*-Quinols: Additive Effects of Copper-Catalyzed Addition of Aryl Boronic Acid to Benzoquinones

**DOI:** 10.3390/ijms24021623

**Published:** 2023-01-13

**Authors:** Dominik Koszelewski, Paweł Kowalczyk, Jan Samsonowicz-Górski, Anastasiia Hrunyk, Anna Brodzka, Justyna Łęcka, Karol Kramkowski, Ryszard Ostaszewski

**Affiliations:** 1Institute of Organic Chemistry, Polish Academy of Sciences, Kasprzaka 44/52, 01-224 Warsaw, Poland; 2Department of Animal Nutrition, The Kielanowski Institute of Animal Physiology and Nutrition, Polish Academy of Sciences, Instytucka 3, 05-110 Jabłonna, Poland; 3Department of Physical Chemistry, Medical University of Bialystok, Kilińskiego 1 Str., 15-089 Białystok, Poland

**Keywords:** *p*-quinols, chemoselectivity, copper-catalyzed addition, polyvinylpyrrolidone (PVP), copper metal–organic frameworks (Cu-MOF), additives, antimicrobial activity, *E. coli* cells, MIC

## Abstract

A mild and efficient protocol for the synthesis of *p*-quinols under aqueous conditions was developed. The pivotal role of additives in the copper-catalyzed addition of aryl boronic and heteroaryl boronic acids to benzoquinones was observed. It was found that polyvinylpyrrolidone (PVP) was the most efficient additive used for the studied reaction. The noteworthy advantages of this procedure include its broad substrate scope, high yields up to 91%, atom economy, and usage of readily available starting materials. Another benefit of this method is the reusability of the catalytic system up to four times. Further, the obtained *p*-quinols were characterized on the basis of their antimicrobial activities against *E. coli*. Antimicrobial activity was further compared with the corresponding 4-benzoquinones and 4-hydroquinones. Among tested compounds, seven derivatives showed an antimicrobial activity profile similar to that observed for commonly used antibiotics such as ciprofloxacin, bleomycin, and cloxacillin. In addition, the obtained *p*-quinols constitute a suitable platform for further modifications, allowing for a convenient change in their biological activity profile.

## 1. Introduction

*p*-Quinol skeletons are frequently found in many bioactive natural products [1,2,3,4], and they also serve as useful synthetic building blocks [5,6,7,8,9,10,11,12,13,14,15,16,17,18,19,20,21,22,23,24,25,26,27] (Figure 1). Moreover, *p*-quinol glycosides are known from analgesic activities [27]. 

The examination of the toxic effect of 1,4-cyclohexadienones on bacterial cells can provide appropriate antimicrobial agents against microbial clinical pathogens [28,29,30,31,32,33] (Figure 1). The aim of the present study is the development of an efficient method of isolating *p*-quinol derivatives with aryl and heteroaryl groups and their validation against model pathogenic strains of *Escherichia coli* K12 (with native lipopolysaccharide (LPS) in its structure) and R2–R4 (LPS of different lengths in its structure). General methods of obtaining *p*-quinols are based on the dearomatization of para-substituted phenols via oxidation using hypervalent iodine reagents [34]. However, this approach often suffers from low yields because of competitive oligomerization, especially in the case of oxidation of 4-arylphenols [35]. Thus, the development of a new method to overcome these limitations is of great importance [35,36,37,38,39].

## 2. Results and Discussion

### 2.1. Chemistry

Recently, we have developed a sustainable method for the synthesis of *p*-quinols based on the copper-catalyzed addition of phenylboronic acid to quinone, leading to a target product formation under aqueous conditions [40]. However, desired products were obtained with moderate yields. Additionally, the formation of side products hampered product purification. Therefore, the previously developed method is characterized by low atom economy [41,42,43,44,45,46,47]. 

As a continuation of our research on the search for new catalytic activities of copper salts, we focused our efforts on elaborating an efficient and sustainable method of obtaining desired *p*-quinols (Figure 1). Based on our recent findings regarding the activity of copper (I) iodide [40], the model addition reaction of phenylboronic acid (1 mmol) and benzoquinone (1 mmol) was conducted in distilled water at 20 °C under atmospheric pressure (Figure 1, Table 1, entry 1). As a result, the mixture of products **1** and **15** with 51% and 9% yields, respectively (Figure 1, Table 1, entry 1), was obtained. In order to discover more reusable catalysts and to enhance the reaction efficiency, solid supported catalysts have been developed for catalytic applications [48]. In addition to this, Cu(I) species immobilized onto various supports, such as silica [49], zeolites [50], activated charcoal [51], and amine functionalized polymers [52] have been reported recently. The character of supporting materials on which nanoparticles are stabilized plays a crucial role in catalysis as it provides a highly active catalyst surface, which increases the reaction rate. Inspired by the work of Liu et al. [53], an anion exchange resin (Amberlite IRA 400) was employed as the additive, resulting in target product **1** with an enhanced yield (Table 1, entry 2). Encouraged by this result, various different adsorbents including ionic polymers having quaternary ammonium were tested (Table 1). Further improvement in the reaction yield was achieved by the application of montmorillonite, producing target product 1 with 64% yield. It should be mention that the formation of 1,4-addition product **15** was not observed (Table 1, entry 3). An application of basic amberylst [54] resulted in the formation of target product **1** with a reduced reaction yield (18%, Table 1, entry 4), while the application of quaternary ammonium based Dowex-1 provided *p*-quinol **1** with 68% (Table 1, entry 5). Ionic polymers like Dowex-1 were already found to be efficient support for CuI catalysts in Huisgen’s 1,3-dipolar cycloaddition [55]. No impact on the reaction yield was observed in the case of using silica gel or aluminum oxide (Table 1, entries 6 and 7). Chavan and his group demonstrated cellulose supported cuprous iodide as an efficient catalyst in the click synthesis of 1,4-disubstituted 1,2,3-triazoles [56]; however, we have not observed any pivotal impact of this additive on the studied reaction (Table 1, entry 8). 

Metal organic frameworks (MOFs) have been effectively used as heterogeneous catalysts improving efficiency and selectivity of various reactions. The specific porous structure of MOF containing organic and inorganic active sites is a useful and effective alternative to heterogeneous catalysts [57,58]. Two MOFs prepared in accordance with the literature procedures [59,60] were tested in the model reaction. However, a product was obtained with a moderate yield up to 42% (Table 1, entries 9 and 10). On the other hand, the advantage of MOFs was their easy separation from the reaction mixture and reusability. The MOF-1 catalyst was used three times. The yield after the third cycle was 27%. The catalyst was isolated by filtration on a silica sinter. The decrease in yield could be related to the physical loss of the catalyst during separation from the reaction mixture. Colloidal synthesis offers a route to nanoparticles (NPs) with controlled composition and structural features. Polyvinylpyrrolidone (PVP) can serve as a surface stabilizer, growth modifier, nanoparticle dispersant, and reducing agent [61]. High surface-to-volume ratios make metal colloids promising candidates for active catalysts [62]. Copper-PVP composites have been found to be efficient catalysts for click reactions [63,64,65]. Among three different PVPs, an application of this with 3500 average molecular weight provided target product **1** with 84% yield. Moreover, for each type of used PVP, only a formation of 1,2-addition product **1** was observed (Table 1, entries 11–13). An increased amount of the used PVP, 15 mol% and 20 mol%, did not affect the reaction yield (Table 1, entries 14 and 15). Next, the impact of temperature on the model reaction was studied. When the reaction was conducted at 30 °C, the yield increased to 89%. However, the further elevation of temperature led to a decrease in yield, which may be explained by the changes in the colloidal structure of the catalyst (Table 1, entries 16-17). The application of methanol as a reaction medium, which was found previously [40] to be suitable for the studied reaction, resulted in target product **1** with reduced yield (Table 1, entry 18, Figure 2).

The model reaction was carried out under the optimized procedure. Formed product 1 was separated by extraction with EtOAc, followed by purification using column chromatography. The remaining aqueous phase containing the CuI-PVP catalytic system was employed for another run with the fresh portion of substrates (Figure 3). Due to the possibility of repeated use of the reaction medium containing the catalyst, the *E*-factor for the developed procedure is lower than that for the reaction with copper iodide alone. 

Finally, the elaborated protocol was applied for the synthesis of the series of *p*-quinols **2**–**10** with good to very high yields for various boronic acids (Figure 4). In case of using 2-methyl-1,4-benzoquinone as the substrate, only one of two possible regioisomers was obtained with 82% (*p*-quinol **10**). The developed protocols were also revealed to be efficient for heterocyclic boronic acids, resulting in the formation of products **8** and **9** with good yield. Principally, with the exception of compound 6, all others were obtained with much higher yields compared to the reaction carried out under conditions without PVP presented in the previous work [40].

### 2.2. Cytotoxic Studies of the Library of p-Quinols ***1***–***10***, and Parent Benzo- and Hydroquinones ***11***–***14***

The toxic effect on bacterial cells was studied after the analysis of the MIC and MBC test for all 14 tested compounds (Figure 4 and Figure 5). The MIC values were observed in the range of 0.2–1.4 µg/mL and 2–82 µg/mL for MBC values in the analyzed model strains K12, R2, R3, and R4 (Figure 6 and Figure 7), which had specific functional groups in the structure of the 4-hydroxycyclohexa-2,5-dienones.

### 2.3. Analysis of Bacterial DNA Isolated from E. coli R2–R4 Strains Modified with Tested p-Quinols 

The obtained MIC values, as well as our previous studies with various types of the analyzed compounds [65,66,67,68,69,70,71,72,73,74,75,76,77,78,79,80], (Figure 8) indicate that *p*-quinols show a strong toxic effect on the analyzed bacterial model strains. Based on the MIC and MBC values, the analyzed compounds **5**, **7**, **10,** and **11** were selected for further analyses (on the basis of their highest biological activity similar to that of antibiotics) and their values were selected for further studies related to the analysis of oxidative stress in the cell by modifying them with the bacterial DNA obtained from the analyzed strains. On the other hand, compounds numbered **1**–**4**, **6**, **8**, **9,** and **12**–**14** showed higher activity than selected compounds **5**, **7**, **10**, and **11,** with activity similar to the biological activity of antibiotics (see Appendix A, Table 2).

The conducted research proved that the analyzed and newly synthesized compounds have the potential (further functionalization) to find a new innovative application in the future after their more in-depth examination on e.g. specific cell cultures as potential “replacements” of currently used antibiotics commonly used in hospital and clinical infections (Figure 9). 

It is noteworthy that both the hydrophilic compound 5 containing two methoxy groups in its structure as well as the lipophilic *p*-quinol 7 show the highest antimicrobial activity. This may indicate a different mechanism of action of these compounds on selected strains of *E. coli* (Figure 10 and Figure 11)A significant effect of the methyl group present in the 2-position of the *p*-quinol ring on the increased antimicrobial activity was also noted [67,68,69,70,71,72,73,74,75,76,77,78,79,80] (Figure 4). Dysfunction of bacterial membranes containing different lengths of LPS in model bacterial strains is an ideal model to assess the effectiveness of these compounds in relation to the antibiotics used [67,68,69,70,71,72,73,74,75,76].

## 3. Materials and Methods

### 3.1. Microorganisms and Media

The entire methodology and all materials and media used are detailed in previous work [67,68,69,70,71,72,73,74,75,76], and data were analyzed by the Tukey test indicated by (*p* < 0.05): * *p* < 0.05, ** *p* < 0.1, *** *p* < 0.01.

### 3.2. Minimum Inhibitory Concentration (MIC) and Minimum Bactericidal Concentration (MBC)

The MIC was estimated by a microtiter plate method using sterile 48 or 96-well plates [69,70,71,72,73,74,75,76,77,78]. The mother liquor was prepared in DMSO at a concentration of 20 mg/mL^−1^. Samples at a given concentration were prepared by diluting the mother liquor with distilled water. 

### 3.3. Chemicals

The chemistry used for the research came from Sigma-Aldrich, Saint Louis, MI, USA.

### 3.4. General Procedure for the Synthesis of p-Quinols

Quinone derivative (0.4 mmol), boronic acid derivative (0.4 mmol), and catalyst (10 mol%) together with PVP (10 mol%) were stirred in distilled water (2 mL) at room temperature. 

**4-Hydroxy-4-phenyl-cyclohexa-2,5-dienone (1)**. Compound **1** was obtained according to the general method with 89% yield (166 mg, 0.89 mmol) as a white solid; m.p. 103–104 °C [Lit. m.p. 102–103 °C; [81]; ^1^H NMR (400 MHz, CDCl_3_) *δ* 7.55–7.43 (m, 2H), 7.43–7.28 (m, 3H), 6.90 (d, *J* = 10.1 Hz, 2H), 6.22 (d, *J* = 10.1 Hz, 2H), 2.71 (s, 1H); ^13^C NMR (100 MHz, CDCl_3_) *δ* 185.7, 150.8, 138.7, 128.9, 128.4, 126.8, 125.2, 71.0. NMR data were in accordance with those reported in the literature [82].

**4-Hydroxy-4-(4′-methyl)-phenyl-cyclohexa-2,5-dienone (2)**. Compound **2** was obtained according to the general method with 72% yield (144 mg, 0.72 mmol) as a white solid; m.p. 134–137 °C, Lit. m.p. 134–137 °C [83] ^1^H NMR (400 MHz, CDCl_3_) δ 7.36 (d, *J* = 8.3 Hz, 2H), 7.21–7.13 (m, 2H), 6.88 (d, *J* = 10.0 Hz, 2H), 6.20 (d, *J* = 10.0 Hz, 2H), 2.64 (s, 1H), 2.35 (s, 3H); ^13^C NMR (100 MHz, CDCl_3_) *δ* 185.7, 150.9, 138.3, 135.7, 129.6, 126.7, 125.1, 70.9, 21.0. NMR data were in accordance with those reported in the literature [84].

**4-Hydroxy-4-(4′-chloro)-phenyl-cyclohexa-2,5-dienone (3)**. Compound **3** was obtained according to the general method with 64% yield (141 mg, 0.64 mmol) as a white solid; m.p. 170–172 °C, Lit. m.p. 171–172 [85]; ^1^H NMR (400 MHz, CDCl_3_) *δ* 7.41 (d, *J* = 8.7 Hz, 2H), 7.34 (d, *J* = 8.7 Hz, 2H), 6.85 (d, *J* = 10.1 Hz, 2H), 6.22 (d, *J* = 10.1 Hz, 2H), 2.74 (s, 1H); ^13^C NMR (100 MHz, CDCl_3_) *δ* 185.4, 150.3, 137.2, 134.4, 130.4, 127.0, 126.8, 70.6. NMR data were in accordance with those reported in the literature [40].

**4-Hydroxy-4-(4′-formyl)-phenyl-cyclohexa-2,5-dienone (4)**. Compound **4** was obtained according to the general method with 31% yield (66 mg, 0.31 mmol) as a white solid; m.p. 158–159 °C; ^1^H NMR (400 MHz, CDCl_3_) *δ* 10.01 (s, 1H), 7.89 (d, *J* = 8.6 Hz, 2H), 7.66 (d, *J* = 8.3 Hz, 2H), 6.87 (d, *J* = 10.1 Hz, 2H), 6.28 (d, *J* = 10.1 Hz, 2H), 2.84 (s, 1H); ^13^C NMR (100 MHz, CDCl_3_) δ 191.6, 185.2, 149.7, 145.2, 136.2, 130.2, 127.5, 126.1, 71.0. NMR data were in accordance with those reported in the literature [40].

**4-(3,4-Dimethoxyphenyl)-4-hydroxycyclohexa-2,5-dien-1-one (5)**. Compound **5** was obtained according to the general method with 43% yield (106 mg, 0.43 mmol) as a white solid; m.p. 164–166 °C; ^1^H NMR (400 MHz, CDCl_3_) *δ* 7.43 (d, *J* = 8.6 Hz, 2H), 7.11–7.03 (m, 2H), 6.92 (d, *J* = 8.2 Hz, 1H), 6.89 (d, *J* = 8.6 Hz, 2H), 3.93 (s, 3H), 3.91 (s, 3H); ^13^C NMR (100 MHz, CDCl_3_) *δ* 185.2, 154.8, 149.1, 134.0, 128.0, 118.9, 115.6, 111.6, 110.3, 69.7, 56.0, 55.9. NMR data were in accordance with those reported in the literature [40].

**4-Hydroxy-4-(4′-hydroxymethylphenyl)phenyl-cyclohexa-2,5-dienone (6)**. Compound **6** was obtained according to the general method with 29% yield (63 mg, 0.29 mmol) as a white solid; m.p. 181–182 °C; ^1^H NMR (500 MHz, Acetone-d_6_) *δ* 8.20 (s, 1H), 7.30 (d, *J* = 8.6 Hz, 2H), 6.92–6.78 (m, 6H), 4.57 (s, 2H); 13C NMR (126 MHz, Acetone) *δ* 186.2, 158.6, 154.5, 150.2, 137.3, 128.9, 121.5, 117.8, 117.0, 69.9, 64.2. NMR data were in accordance with those reported in the literature [40].

**4-Hydroxy-4-(4′-biphenyl)phenyl-cyclohexa-2,5-dienone (7)**. Compound **7** was obtained according to the general method with 54% yield (142 mg, 0.54 mmol) as a white solid; m.p. 146–148 °C; ^1^H NMR (400 MHz, CDCl_3_) *δ* 7.59–7.48 (m, 4H), 7.48–7.37 (m, 2H), 7.37–7.27 (m, 1H), 7.03–6.93 (m, 3H), 6.84 (d, *J* = 8.9 Hz, 2H), 4.80 (s, 1H); ^13^C NMR (101 MHz, CDCl_3_) *δ* 185.0, 158.0, 151.8, 150.2, 140.6, 135.6, 128.7, 128.3, 126.9, 126.8, 121.0, 117.8, 116.4, 70.4. NMR data were in accordance with those reported in the literature [86].

**4-Hydroxy-4-(thiophen-3-yl) phenyl-cyclohexa-2,5-dienone (8)**. Compound **8** was obtained according to the general method with 76% yield (204 mg, 0.76 mmol) as a white solid; m.p. 138–139 °C; ^1^H NMR (400 MHz, CDCl_3_) *δ* 7.41–7.21 (m, 2H), 7.09–6.89 (m, 3H), 6.26–6.04 (m, 2H), 3.64 (s, 1H); ^13^C NMR (100 MHz, CDCl_3_) *δ* 185.9, 150.7, 140.3, 126.9, 126.5, 125.2, 122.0, 69.5.; HRMS calculated for C_10_H_8_O_2_S [M+H]^+^: 193.0323, found: 193.0320.

**4-Hydroxy-4-(furan-3-yl) phenyl-cyclohexa-2,5-dienone (9)**. Compound **9** was obtained according to the general method with 69% yield (174 mg, 0.69 mmol) as a white solid; m.p. 131–133 °C; ^1^H NMR (400 MHz, CDCl_3_) *δ* 7.44 (d, *J* = 24.8 Hz, 2H), 6.95 (d, *J* = 10.1 Hz, 2H), 6.34 (d, *J* = 1.0 Hz, 1H), 6.19 (d, *J* = 10.2 Hz, 2H), 2.66 (s, 1H); ^13^C NMR (100 MHz, CDCl_3_) *δ* 149.7, 143.9, 139.6, 127.0, 125.8, 108.1, 66.8.; HRMS calculated for C_10_H_8_O_3_ [M+H]^+^: 177.0552, found: 177.0555.

**4-Hydroxy-2-methyl-4-phenyl-2,5-cyclohexadienone (10)**. Compound **10** was obtained according to the general method with 82% yield (164 mg, 0.82 mmol) as a white solid; m.p. 76–77 °C [Lit. m.p. 73–75 °C (diethyl ether, hexane) [86]; ^1^H NMR (400 MHz, CDCl_3_) δ 7.47–7.25 (m, 5H), 6.87–6.77 (m, 1H), 6.19–6.03 (m, 2H), 2.99 (s, 1H), 1.83 (s, 3H); ^13^C NMR (100 MHz, CDCl_3_) *δ* 186.7, 161.7, 151.9, 138.5, 128.7, 128.0, 126.3, 125.7, 125.2, 73.2, 18.4. NMR data were in accordance with those reported in the literature [87].

**2-Phenyl-1,4-benzoquinone (15)**. White solid; m.p. 118–119 °C [Lit. m.p. 117–118 [88]; ^1^H NMR (400 MHz, CDCl_3_) δ 7.55–7.38 (m, 5H), 6.90–6.78 (m, 3H); ^13^C NMR (100 MHz, CDCl_3_) *δ* 187.5, 186.6, 145.9, 137.0, 136.2, 132.6, 130.1, 129.2, 128.5. NMR data were in accordance with those reported in the literature [88].

### 3.5. Preparation of Copper–PVP Colloids in Water

Colloidal solution of copper PVP was prepared by vigorously stirring for 30 minutes at room temperature a mixture of CuI (8 mg, 0.04 mmol) and PVP (10%mol) in distilled water (4 mL). 

## 4. Conclusions

The novel efficient protocol for obtaining *p*-Quinols was developed. The crucial role of additives in the reaction course was shown. The pivotal role of polyvinylpyrrolidone (PVP) in a colloidal catalyst system was revealed, and it was applied for the synthesis of various *p*-Quinols with very good yields and excellent chemo- and regioselctivity. Moreover, the developed colloidal system can be reused several times, which significantly reduces the overall cost of the synthesis. 

## Data Availability

On request of those interested.

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
