# Peer review of "Synthesis and Antimicrobial Activity of the Pathogenic E. coli Strains of p-Quinols: Additive Effects of Copper-Catalyzed Addition of Aryl Boronic Acid to Benzoquinones"

_ijms, 2023, doi:10.3390/ijms24021623_

Round 1

Reviewer 1 Report

The manuscript described a Cu-catalyzed 1,4-benzoquinones synthesis and their antimicrobial activity. The synthetic method presents a high chemoselectivity in a mild condition, the substates compatibility and corresponding antimicrobial activity is good as well. The overall academic significance should be sufficient to be published in IJMS, after some minor issues are addressed.

1.The abstract should be written in a briefer manner, introducing for the most significant impact of the research should be sufficient.

2.Similarly, the description of the condition optimization (from line 105-163) should be shorten and refined as well, since there is no need to discuss each background of the additive in detail.

3.Table 1 and Figure 4 should add the corresponding chemical reaction equation.

4.line 19, the “polivinylopirolidone” should be “polyvinylpyrrolidone”;

Author Response

RESPONSE TO REVIEWERS 1

               Firstly, we would like to express our gratitude to Reviewer 1 for their suggestions that allowed us to considerably improve our manuscript. We have revised the text according to the suggestions and we hope that you will now find it suitable for publication in the International Journal of Molecular Sciences. Below, please find the detailed information on the changes in the manuscript with answers to all comments. All changes made in the manuscript were marked up using the “Track Changes” function.

Reviewer 1: The manuscript described a Cu-catalyzed 1,4-benzoquinones synthesis and their antimicrobial activity. The synthetic method presents a high chemoselectivity in a mild condition, the substates compatibility and corresponding antimicrobial activity is good as well. The overall academic significance should be sufficient to be published in IJMS, after some minor issues are addressed.

Response: We are very grateful to the reviewer for the effort put into reviewing our work and for the comments that will improve its quality.

Reviewer 1:1.The abstract should be written in a briefer manner, introducing for the most significant impact of the research should be sufficient.

Response: We are grateful for this remark. As suggested by the Reviewer, the abstract was revised and modified

Reviewer 1: 2.Similarly, the description of the condition optimization (from line 105-163) should be shorten and refined as well, since there is no need to discuss each background of the additive in detail.

Response: We are grateful for this remark. However, due to the different nature and characteristics of the additives used, we consider it practical to include a detailed description of the results obtained in order to facilitate the potential reader's analysis of their impact on the course of the tested reaction.

Reviewer 1: 3.Table 1 and Figure 4 should add the corresponding chemical reaction equation.

Response: We are grateful for this remark. As suggested by the Reviewer, reaction equation was provided to the Table 1 and to the Figure 4

Reviewer 1: 4.line 19, the “polivinylopirolidone” should be “polyvinylpyrrolidone”;

Response: We are grateful for this remark. As suggested by the Reviewer, the name of the substance has been corrected

Author Response

RESPONSE TO REVIEWERS 2

               Firstly, we would like to express our gratitude to Reviewer 1 for their suggestions that allowed us to considerably improve our manuscript. We have revised the text according to the suggestions and we hope that you will now find it suitable for publication in the International Journal of Molecular Sciences. Below, please find the detailed information on the changes in the manuscript with answers to all comments. All changes made in the manuscript were marked up using the “Track Changes” function.

Reviewer 2: The manuscript of Koszelewski et al entitled “Synthesis and antimicrobial activity of pquinols. Additive effects on copper-catalyzed addition of aryl boronic acid to benzoquinones” describes the development of a new protocol to synthesize p-quinols followed by biological application of the products against pathogenic E. coli strains. After careful analysis, I recommend the author consider the points described below.

Response: We are very grateful to the reviewer for the effort put into reviewing our work and for the comments that will improve its quality.

Reviewer 2: 1. I suggest authors include in the title that antimicrobial activity refers to pathogenic E. coli strains.

Response: We are grateful for this remark. According to the Reviewer suggestion the title was modified.

Reviewer 2: 2. Although I am not a native English speaker, seems to me that in lines 56-57 the sentence is not clear. Could the authors check it?

Response: We are grateful for this remark. According to the Reviewer suggestion mentioned sentence was revised and corrected.

Reviewer 2: 3. In line 63 the authors introduced the abbreviation LPS at the first time in the manuscript. I think it refers to “lipopolysaccharide”. Could the authors add what LPS means? Although this is a widely used term in the area, it is important to always pay attention to the interdisciplinary of the manuscript. Readers from different areas may be interested in this text.

Response: We are grateful for this remark. According to the Reviewer suggestion we have included the meaning of the abbreviation LPS.

Reviewer 2: 4. In scheme 1, for compound 15, please, inform that R2 = Ph. In this scheme, I suggest also inverting the structure of “1-10” to be the same as all the rest of the manuscript.

Response: We are grateful for this remark. According to the Reviewer suggestion the Scheme 1 was revised and supplemented with R2=Ph for compound 15. The structure of “1-10” was inverted

Reviewer 2: 5. In line 143, please, check the capital letter in “. two MOFs …”.

Response: We are grateful for this remark. It was corrected

Reviewer 2: 6. In lines 158-159, “Increased amount of the used PVP up to 20%mole did not increase the reaction yield”, I recommend also informing the amount 15% mol, once “(Table 1, entries 14 and 15)” was mentioned.

Response: We are grateful for this remark. According to the Reviewer suggestion we have included the amount of 15% mol.

Reviewer 2: 7. Analyzing the discussion of polyvinylpyrrolidone (PVP) (lines 151-155), Dynamic light scattering (DLS) (lines 164-174), and “General Procedure for the Synthesis of pquinols” (lines 386-387) it is not clear to me if CuI was used as NP or as a traditional catalyst that becomes a NP in the reaction. If CuI was applied as a NP, how was it prepared? In line 184 “copper particles” is mentioned. Are these refer to Cu, CuI, or CuIPVP or is it a general term? In General Procedures the authors only say “catalyst (10 mol%) together with PVP (10 mol%)”. Could the authors make this clear in the discussion and procedure?

Response: We are grateful for this remark. According to the Reviewer suggestion we have provided procedure for copper particles preparation. As can be seen from DLS measurements the size distribution of the obtained particles is larger than that of the nanoparticles. We conclude that these are colloidal systems.

Reviewer 2: 8. Please, check in 183, if “extraction” should be used instead of “excretion”.

Response: We are grateful for this remark. It was corrected

Reviewer 2: 9. In lines 194-198, could the authors improve the discussion? For instance, describe which one provides a better yield, the effect of substituents in the reaction yield, comment on the use of heterocycle, compare with previous work

Response: We are grateful for this remark. According to the Reviewer suggestion, the discussion regarding obtained results was revised and improved.

Reviewer 2: 10. In lines 195-197 the authors say “In case of using 2-methyl-1,4-benzoquinone as the substrate only one from two possible isomers was obtained with 82% (p-quinol 10).” The possible isomer probably is an enantiomer. Could the authors inform in the text all was confirmed the formation of only one enantiomer? Is there a possible explanation for this result?

Response: We are grateful for this remark. In the case of compound number ten, we meant the second regioisomer, 4-hydroxy-3-methyl-4-phenyl-2,5-cyclohexadienone, which can be formed and which is not formed under the developed conditions. The resulting compound 10 is chiral but racemic.

Reviewer 2:

  1. Could the authors also add a table with all values related to figures 6,7 and 8. As there are not many compounds, maybe it would be interesting to also add a table. These figures are important for comparative visual analysis, but the exact values for each case would also be important.

In our opinion, we did not include the table with the results themselves, because we decided that they would be more readable in the form of graphs presented in the above-mentioned figures. The more so that we have selected compounds with further functionalization. A similar presentation of the results was used in our previous works.

  1. Please, correct the number “10-3” in line 235.

the value 10-3 has been corrected and enclosed like other values in parentheses. We apologize for this oversight. The changes is marked in yellow.

  1. I suggest the authors rethink the sentence “Performed studies proved that the analyzed and newly synthesized compounds can potentially be used as "substitutes" for the currently used antibiotics in hospital and clinical infections.” in lines 269-271. Although it is clear that the authors are discussing a potentiality, I understand that these results may not be enough to "prove" that these compounds can be used as " substitutes " for "antibiotics in hospital and clinical infections". Perhaps the authors could highlight the potential of these compounds, saying that more studies could be carried out in the future, to apply them as antibiotics, i.e. more generally.

The sentence as suggested has been worded differently and an example of it is given below. The changes is marked in yellow.

The conducted research proved that the analyzed and newly synthesized compounds have the potential (further functionalization) to find a new innovative application in the future after their more in-depth examination on e.g. specific cell cultures as potential "replacements" of currently used antibiotics commonly used in hospital and clinical infections.

Reviewer 2: 14. In lines 299-230, please, confirm if the mentioned “table 1” is correct.

Response: We are grateful for this remark. As suggested by the Reviewer, the mentioned part of manuscript was revised and corrected.

Reviewer 2: 15. In lines 301-303, the authors say “…indicate that also p-quinols show a strong cytotoxic effect of the analyzed model bacterial strains K12 and R2 - R4, due to the type and kind of group at para-position.” Could the authors discuss more about it? Which group and type are related to the strong cytotoxic? In my opinion, this is a very important topic not only in this manuscript but also in the field of research.

Response: We are grateful for this remark. As suggested by the Reviewer, the description on the influence of the structure of the tested compounds on their antimicrobial activity was supplemented.

Reviewer 2: 16. Please, check the NMR description of compound 5. Seems to me that there are more aromatic H than expected. Furthermore, confirm if in “6.92 (d, J = 8.2 Hz, 1H)” should not be 2H. In the NMR description of compound 7, seems to me there is an H missing. Just in case, all 1H and 13C should be checked.

Response: We are grateful for this remark. All NMR spectra were carefully revised.
